# Top Canopy Height and Stem Size Variation Enhance Aboveground Biomass across Spatial Scales in Seasonal Tropical Forests

**DOI:** 10.3390/plants12061343

**Published:** 2023-03-16

**Authors:** Zhenhua Sun, Arunkamon Sonsuthi, Tommaso Jucker, Arshad Ali, Min Cao, Feng Liu, Guanghong Cao, Tianyu Hu, Qin Ma, Qinghua Guo, Luxiang Lin

**Affiliations:** 1CAS Key Laboratory of Tropical Forest Ecology, Xishuangbanna Tropical Botanical Garden, Chinese Academy of Sciences, Mengla 666303, China; 2Center of Plant Ecology, Core Botanical Gardens, Chinese Academy of Sciences, Mengla 666303, China; 3University of Chinese Academy of Sciences, Beijing 100049, China; 4School of Biological Sciences, University of Bristol, Bristol BS8 1QU, UK; 5Forest Ecology Research Group, College of Life Sciences, Hebei University, Baoding 071002, China; 6Yunnan Academy of Forestry and Grassland, Kunming 650201, China; 7Administration Bureau of Naban River Watershed National Nature Reserve, Jinghong 666100, China; 8State Key Laboratory of Vegetation and Environmental Change, Institute of Botany, Chinese Academy of Sciences, Beijing 100093, China; 9Institute of Ecology, College of Urban and Environmental Science, Peking University, Beijing 100871, China

**Keywords:** aboveground biomass (AGB), stand structural attributes, UAV LiDAR-based canopy structure, mean top of canopy height (TCH), tropical forest

## Abstract

Tropical forests are biologically diverse and structurally complex ecosystems that can store a large quantity of carbon and support a great variety of plant and animal species. However, tropical forest structure can vary dramatically within seemingly homogeneous landscapes due to subtle changes in topography, soil fertility, species composition and past disturbances. Although numerous studies have reported the effects of field-based stand structure attributes on aboveground biomass (AGB) in tropical forests, the relative effects and contributions of UAV LiDAR-based canopy structure and ground-based stand structural attributes in shaping AGB remain unclear. Here, we hypothesize that mean top-of-canopy height (TCH) enhances AGB directly and indirectly via species richness and horizontal stand structural attributes, but these positive relationships are stronger at a larger spatial scale. We used a combined approach of field inventory and LiDAR-based remote sensing to explore how stand structural attributes (stem abundance, size variation and TCH) and tree species richness affect AGB along an elevational gradient in tropical forests at two spatial scales, i.e., 20 m × 20 m (small scale), and 50 m × 50 m (large scale) in southwest China. Specifically, we used structural equation models to test the proposed hypothesis. We found that TCH, stem size variation and abundance were strongly positively associated with AGB at both spatial scales, in addition to which increasing TCH led to greater AGB indirectly through increased stem size variation. Species richness had negative to negligible influences on AGB, but species richness increased with increasing stem abundance at both spatial scales. Our results suggest that light capture and use, modulated by stand structure, are key to promoting high AGB stocks in tropical forests. Thus, we argue that both horizontal and vertical stand structures are important for shaping AGB, but the relative contributions vary across spatial scales in tropical forests. Importantly, our results highlight the importance of including vertical forest stand attributes for predicting AGB and carbon sequestration that underpins human wellbeing.

## 1. Introduction

Forests play a critical role in global carbon cycling while conserving terrestrial biodiversity at the same time [1,2]. Among forest biomes, tropical forests are not only sequestering a large amount of carbon in standing aboveground biomass (AGB) but also form dense complex stand structures through higher species richness and tree stem variation [3,4]. Forest stand structural complexity defines how species capture and use available resources through variation in both vertical and horizontal tree sizes within a community [5,6], which in turn may greatly influence AGB [7]. Thus, understanding the ecological mechanisms underlying the relationships between forest stand structural complexity and AGB is critical to predicting how forests will respond to anthropogenic impact as well as to managing forests in the context of global climate changes [8,9].

During the last few decades, the relationships between forest stand structural attributes and forest functioning (i.e., AGB or productivity) have been widely reported but remain highly debated [7]. More specifically, recent studies found a positive relationship between stand structural complexity (i.e., tree stem size variation) and AGB in subtropical and tropical forests [3,10]. The observed relationships in previous studies have been linked to the niche complementary mechanisms, which assumes that higher variation in tree stem size variation could lead to an increased resource-use complementarity by allowing the formation of multiple leaf layers and/or a highly packed canopy [5,11]. Nevertheless, while resource-use complementarity may explain the positive relationship between forest diversity, structure and function [12], forest structural complexity will not necessarily always lead to greater AGB due to asymmetric competition for light [7]. Thus, the interplay between stand structural attributes (e.g., tree stem size variation, canopy height, species richness and stem abundance) matters in explaining variation in AGB across tropical landscapes. For example, species richness could indirectly affect AGB through the mediation of tree stem size variation and/or stem abundance, or vice versa [13]. Thus, it has been reported that species richness increases AGB indirectly via promoting stem abundance and/or tree stem size variation in tropical forests [3,14]. Alternatively, the selection hypothesis suggests that the presence of a few productive or highly functioning species may contribute to AGB better than other species within a community [15]. For example, it has been reported that the presence of few large trees may overrule the effects of forest stand structural complexity on AGB in forest communities [16,17].

Among forest stand structural attributes, mean top canopy height (TCH) has been recognized as one of the robust predictors of AGB in tropical forests due to the spatial crown variability in coexisting tree species within a community [18]. However, different forest communities at a local scale could have similar TCH but they may differ in tree stem abundance, stem size variation, and species richness [18,19]. This implies that including information from the canopy should improve our understanding of the relationships between stand structural complexity and AGB in forests. Yet, while horizontal forest stand structure attributes (e.g., tree stem size variation in diameters) have extensively been studied to explain AGB, we lack a complete picture of how vertical stand structural attributes (e.g., TCH) covary with horizontal attributes, and how they together shape AGB in tropical forests. A key challenge is that traditional forest inventory is a ground-based approach, lacking a full assessment of TCH in forests. In this regard, airborne laser scanning (i.e., LiDAR) allows us to measure the forest biophysical parameters at high spatial resolution [9]. Previous studies have shown that LiDAR-based canopy structural attributes are the key determinants of AGB in forests [20,21].

Forest stand structural attributes can vary considerably across tropical landscapes due to subtle changes in topography, soil nutrients and past disturbance, which in turn drives local variation in AGB [22,23,24]. At fine scales, topography affects microclimatic and soil nutrient availability which could have both direct and indirect effects on AGB [25,26,27,28]. For example, thermal and hydrological variations could control tree species abundance and spatial distribution [24,29,30]), which could further shape tree size distribution, leaf trait variation and leaf spatial arrangement [31]. Moreover, nutrient-rich soils could lead to higher plant growth but may also lead to higher plant mortality rates due to species’ competition for resources, which in turn could shape the stem abundance and stem size variation within a community [16,32], thereby shaping AGB directly and indirectly via forest stand structural complexity [33].

In this study, we used tropical forest inventory data at two spatial scales, i.e., 20 m × 20 m (small scale), and 50 m × 50 m (large scale) in southwest China for the purpose of determining the effects of both horizontal and vertical stand structural attributes on AGB across spatial scales while considering the direct and indirect effects of topography. By using a conceptual model (Figure 1), we ask the following research questions. (1) How does TCH affect tree species richness, stem abundance, and stem size variation directly, and how do they together influence AGB directly and indirectly? (2) How does topography affect AGB directly and indirectly via stand structural complexity attributes? (3) What is the relative contribution of stand structural complexity attributes and topography to AGB, and what is the main direct driver of AGB? (4) Do the relationships of AGB with stand structural complexity attributes and topography vary across spatial scales? We hypothesize that TCH enhances AGB directly and indirectly via species richness and horizontal stand structural attributes, but these positive relationships are stronger at a larger spatial scale.

## 2. Materials and Methods

### 2.1. The Study Sites and Forest Plots

This study was conducted in the tropical seasonal rain forests of Yunnan Province located in southwestern China. We collected data from two forest dynamic plots (each plot size = 20-ha, Figure 2), namely, the Nabanhe plot (NBH; 100.601° N, 22.249° E) and Xishuangbanna plots (XSBN; 101.574° N, 21.611° E), which were established according to the standard guidelines issued by the ForestGEO network (http://www.forestgeo.si.edu/, accessed data 10 August 2021). Each forest plot was subdivided into non-overlapping quadrats at two spatial scales: 20 m × 20 m (500 quadrats) and 50 m × 50 m (80 quadrats) which allow us to account for the possible scale-dependence of forest structural patterns and processes and to test whether scale matters in the relationships amongst species diversity, stand structure and AGB. Both forest plots are formed under similar climatic and geographic conditions [34,35].

### 2.2. Forest Inventory and Quantification of Variables

In each plot, all freestanding woody stems with a diameter at breast height (DBH) ≥ 1 cm were identified to species, tagged, measured, and mapped. We used the latest forest inventory data (censused in 2017) and measured species’ woody density values to calculate AGB for each tree, using the pantropical biomass allometric equation (Equation (1)) [36] in the BIOMASS package [37]. The AGB values across individual trees within each quadrat were summed and scaled up to Mg/ha. Species-level wood density was measured by collecting wood core samples from 3–5 individuals per species, following the standard measurement protocols in both field and laboratory [38].
(1)$$\mathrm{AGB}=\exp\left(-2.024-0.896\times E+0.920\times \ln(\mathrm{WD}\right)+2.795\times \ln\left(\mathrm{DBH}\right)-0.0461\times (\ln{\left(\mathrm{DBH}\right)}^{2}))$$
where DBH is the diameter at breast height (cm), WD is wood density (g cm^−3^), and E is the environmental stress factor (i.e., 0.336 for our study area).

Within each quadrat, we quantified tree species richness as the observed number of tree species using the vegan package [39]. Stem size variation, as a proxy of horizontal stand structure, was quantified by the coefficient of variation in stem DBHs within a quadrat [40]. For the quantification of the vertical stand structure, we used the UAV LiDAR data which were collected in September 2017 using a Greenvalley International LiHawk system (GreenValley International, Beijing, China). The system is equipped with a RIEGL VUX-1 UAV laser scanner, which has a maximum ranging capability of 1000 m and provides high-speed data acquisition (550 kHz) using a narrow near-infrared laser beam. The collected UAV LiDAR data of each study site were then pre-processed following the same protocol, including denoising, filtering, and normalization. The filtering steps classified ground points and generated a digital terrain model (DTM) from the ground points. An improved progressive triangulated irregular network densification filtering algorithm integrated into LiDAR360 was used to extract ground points [41], and a DTM in 5 m resolution was interpolated using the ordinary kriging method for each study site. From the DTM, we also extracted the mean elevation of each quadrat. Finally, the normalization step was used to remove the influence of terrain elevation on LiDAR point clouds by subtracting the DTM value from the original point height at the corresponding location. Based on the normalized LiDAR point clouds, a canopy height model (CHM) was produced, and we then calculated the vertical stand structural attribute, i.e., mean top-of-canopy height (TCH) as the mean height of pixels making up the surface of the CHM.

### 2.3. Conceptual Model Development and Statistical Analyses

For the development of a conceptual model to test the proposed hypothesis in this study (Figure 1), we assumed that elevation, TCH, stem size variation, stem abundance and species richness shape AGB directly and indirectly via each other at both spatial scales in tropical forests. For the interplay (i.e., the indirect effects of stand structural attributes on AGB) between stand structural attributes, we assumed that higher stem abundance is expected to lead to greater variation in stem size as well as higher species richness [40]. Moreover, stem size variation is expected to shape species coexistence [42]. As such, higher TCH may allow more tree species to coexist through differential light capture and use, which may lead to higher stem size variation with a large number of stems, and/or by forming a densely packed canopy structure [43,44], thereby shaping AGB simultaneously. In addition, to tease apart how topography influences AGB directly and indirectly via species richness and stand structural attributes [9], elevation was included as the exogenous variable in the model (Figure 1). Thus, we tested the conceptual model using the structural equation models (SEMs) across small and large scales, as it allows us to test the direct and indirect pathways in one integrative model [45].

The SEM fit evaluation was determined by using the following statistical parameters [46]: the chi-square test (*p* > 0.05 shows an accepted SEM), the comparative fit index (CFI), the goodness-of-fit index (GFI) (> 0.90 shows a satisfactory SEM fit), and the standardized root mean square residual (SRMR < 0.08 shows SEM fit with less error). To get the best model fit, we excluded the path between TCH and stem abundance, as this relationship was not significant at a small scale whereas it was weakly negative at a large scale. The direct effect was quantified by considering the standardized regression coefficient of the predictor on the response variable, whereas the indirect effect was quantified by multiplying the direct effects of the predictor on the mediator and then on the response variable in one route. The total effect was quantified by summing the direct and indirect effects of predictors on the response variable. In addition, we calculated the relative contributions (in percentage) of predictors to AGB through the ratio of the standardized coefficient of a given predictor to the sum of all coefficients in SEM. SEMs were evaluated using the *lavaan* package [47].

To meet the assumptions of data normality and homoscedasticity [48], all continuous variables including AGB, stand abundance, stem size variation, species richness and TCH were log-transformed and then standardized (by subtracting the variable’s mean and dividing by the standard deviation) prior to statistical analyses. Elevation was transformed between 0 to 1 using the function of (elevation—mean (elevation))/(max(elevation)-min(elevation)). To complement the results from SEMs, we tested bivariate relationships and Pearson correlations amongst tested predictors across spatial scales. All statistical analyses were conducted in R.3.6.0 [49]. Note that, during statistical analyses, we used the combined data from two forest dynamic plots at two different spatial scales, i.e., 1000 quadrats at a small scale and 160 quadrats at a large scale. A summary of variables used in the analyses is provided in Appendix A.

## 3. Results

The tested SEMs had the best fit to the data and explained variation of 76% and 82% of the variance in AGB at 20 m × 20 m and 50 m × 50 m scales, respectively (Figure 3 and Figure 4). At both spatial scales, TCH (β = 0.21 to 0.55), stem size variation (β = 0.71 to 0.44) and stem abundance (β = 0.17 to 0.30) increased AGB directly (Figure 3a and Figure 4a; Appendix A). Species richness possessed a negligible positive effect on AGB at a 20 m × 20 m scale (β = 0.04, Figure 3a, Appendix A) but a negative effect at a 50 m × 50 m scale (β = −0.11, Figure 4a, Appendix A). However, the strength of the positive effects of TCH and stem abundance on AGB increased, whereas the effect of stem size variation decreased on AGB with increasing spatial scale. As such, the negative direct effect of species richness on AGB seemed to be important at a large spatial scale. In addition, elevation increased AGB directly across both scales (β = 0.10 to 0.12, Figure 3a and Figure 4a, Appendix A); however, this effect was a little higher at a larger spatial scale.

Most of the indirect effects on AGB were relatively negligible at both spatial scales. However, TCH possessed a strong positive indirect effect on AGB via stem size variation at a scale of 20 m × 20 m (β = 0.44, Figure 3b, Appendix A) and 50 m × 50 m (β = 0.33, Figure 4b, Appendix A). The indirect effect of stem abundance on AGB via stem size variation was negligible at a scale of 20 m × 20 m (Figure 3b, Appendix A) but negative at a scale of 50 m × 50 m (Figure 4b, Appendix A) due to the divergent direct effects on species richness (β = 0.60, Figure 4b) and stem size variation (β = −0.11, Figure 4b). In addition, stem size variation and abundance promoted species richness at a scale of 20 m × 20 m (β = 0.09 to 0.66, Figure 3b, Appendix A), and the indirect effects on AGB mediated by species richness were negligible. Regarding the indirect effects of elevation, we found negligible effects on AGB (Figure 3b and Figure 4b, Appendix A). However, elevation decreased TCH but increased stem abundance directly at both spatial scales, yet it did not strongly influence species richness and stem size variation (Figure 3a and Figure 4a).

The relative contributions result showed that stem size variation was the most important predictor, followed by TCH, thereby contributing 65% and 28% of the explained variance in AGB at a small scale (Figure 3c). In contrast, TCH was the most important predictor, followed by stem size variation by contributing 49 % and 43% of the explained variance in AGB at a large scale (Figure 4c). These comparative results indicated that horizontal stand structure is relatively important at a small scale whereas both horizontal and vertical stand structures are almost equally important at a large scale. Although TCH promoted stem size variation at both scales, the reverse relationship (i.e., the effect of stem size variation on TCH) might be also true and consistent. 

The bivariate relationships provided support to the tested SEMs where most of the relationships were consistent with the direct effects, as shown in the SEMs (Appendix A). However, we noted a slight positive relationship of species richness with elevation and TCH at 20 m × 20 m, as well as the negative relationship between species richness and TCH that changed to non-significant at a scale of 50 m × 50 m. These small mismatches indicated the necessity of using multiple multivariate analyses for better understanding the complex relationships; for example, species richness was also controlled by many other factors in SEMs.

## 4. Discussion

In this study, we tease apart the direct and indirect effects of TCH, stem size variation, stem abundance and species richness on AGB along elevational gradients across two spatial scales in tropical forests. We found partial support for our proposed hypothesis that TCH, stem size variation and abundance increased AGB as compared to species richness at both small and large spatial scales. These positive relationships of horizontal and vertical stand structural attributes with AGB are indeed due to light capture and use by component species and interacting individual trees within a forest community, and hence supporting the niche complementarity effect [5,50]. However, we did not find a positive effect of species richness on AGB, but rather a negative relationship between species richness and AGB at a larger scale, indicating the selection or competitive exclusion effect [51].

Our results confirm that the positive relationship between stand structural attributes, especially stem size variation and stem abundance, and AGB can be extended to other forest types, as previously shown in boreal [42], temperate [51], and tropical forests [3]. The contribution of our study emphasizes the importance of TCH in ecological models for predicting AGB in tropical forests [52]. Thus, to the best of our knowledge, our study explores the effects of both UAV-LiDAR-derived canopy structural attributes and census-based horizontal stand structural attributes on AGB in tropical forests, which could further enhance our understanding of carbon sequestration that underpins human wellbeing. 

Previously when fine-scale canopy height information was unavailable, the observed strong positive effect of stem size variation on AGB was indirectly ascribed to the higher vertical occupation of available canopy space by various sizes of trees and higher species richness [3,42,51]. Here, we further show that TCH contributed more comprehensively to stem size variation than stem abundance and species richness in tropical forests at both small and large spatial scales. Our findings were not just in accord with early findings in other forests, but also suggest that higher TCH may strengthen the stem size variation by providing more canopy space to fill, which allows more leaf area to intercept light and in turn increases forest productivity [33,44]. The underlying ecological mechanisms appear to be largely dependent on individual plant responses to light availability and crown complementarity among individual stems [5,50,53]. For instance, the growth rate difference between light-intolerant and shade-tolerant species may first define the vertical portioning of canopy height by increasing overall canopy space and then occupy these spaces efficiently through crown complementarity. Besides, we found the effect of TCH on AGB increased with increasing spatial scales, which is consistent with the general notion that large-scale climatic factors related to water and energy balance could shape canopy height and thus influence AGB and carbon sequestration [54].

Counter to the expectation that AGB will weakly increase with species richness [3,55], our study shows that species richness was negatively or negligibly related to AGB across spatial scales. This lack of a significant positive effect of species diversity on AGB might be attributable to the dominance of certain productive species in the studied forests, which might dilute the effect of species diversity [51]. For example, *Parashorea chinensis* was not just an emergent but also the monodominant species of the forest in the XSBN plot, contributing around 22% of the total AGB separately compared to the rest of AGB shared by around 390 other tree species. *P. chinensis* is wind- and gravity-dispersed, and as a result, most of the seeds fall within 10 m of conspecific adults, thereby causing the strongly aggregated distribution pattern in the valley and lower slope [56]. As the selection hypothesis posits that species diversity effects on AGB are more likely driven by the presence of highly productive species or emergent species in diverse communities [15], both the niche complementarity effect and selection effect may together exist as the main mechanisms for shaping AGB in our studied tropical forests across spatial scales. However, we did not find the consistent positive effects of TCH and stem size variation on tree species richness as previously shown for tree [43], liana [57], and different animal groups [58,59]. This result could be due to the reason that higher species richness may occur when TCH is lower, but stem abundance is higher. However, we found that higher stem abundance and lower TCH occurred on higher elevations (ridges) in our studied forests, where seasonal drought may only allow drought-tolerant species to coexist.

Our tested SEMs show that elevation had both direct and indirect effects via stand structural attributes on AGB, but the relative contribution of elevation to AGB was relatively small compared to stand structural attributes. However, we found that TCH and stem abundance rather than species richness and stem size variation mediated the divergent pathways of elevation to AGB, indicating the importance of hydrological controls on forest structural attributes [60]. Moreover, TCH was higher in the gulley of lower elevations than on the ridges, whereas stem abundance increased with increasing elevation, suggesting strong underlying ecological gradients shaped by topography [9]. Thus, despite the fact that the importance of TCH has been linked with AGB, we do acknowledge that including only TCH as a vertical structure attribute versus three horizontal structure attributes is imperfect and further studies are needed to incorporate more proxies of vertical structure [61], e.g., maximum canopy height [62], gap fraction [63], and canopy rugosity [64]. 

One more caveat of this study is that the local habitat is not represented solely based on the elevation. There are other abiotic factors, such as topography-related hydrological feathers [65], soil properties [22,66], and legacies of human impact [67] that impact species diversity and AGB. These contexts are not explicitly included in the present analyses, but they could have potentially influenced the observed variation in forest stand structure attributes and forest biomass. 

## 5. Conclusions

This study shows that TCH (as a proxy of vertical stand structure) and stem size variation (a proxy of horizontal stand structure) boost AGB across spatial scales in tropical forests. The negative to negligible effects of species richness on AGB suggest the competitive exclusion effect, and hence, it is important to test the influence of large trees in future studies. As such, the positive effects of stem abundance on species richness and AGB, the weak effect on stem size variation and the lack of any relationship with TCH could indicate the role of few productive tree species in the studied forests. In addition, the divergent stem abundance, size variation and TCH pathways mediate the influences of topography on AGB, indicating the differential roles of microclimatic conditions on biotic factors. Although we did not test the actual ecological mechanisms through experimental data, the observed results show that light capture and use, modulated by stand structure, seemed to be important for higher AGB, and these effects were stronger at a large scale. Thus, we argue that both horizontal and vertical stand structures are important for shaping AGB, but the relative contributions vary across spatial scales in tropical forests.

## Figures and Tables

**Figure 1 plants-12-01343-f001:**
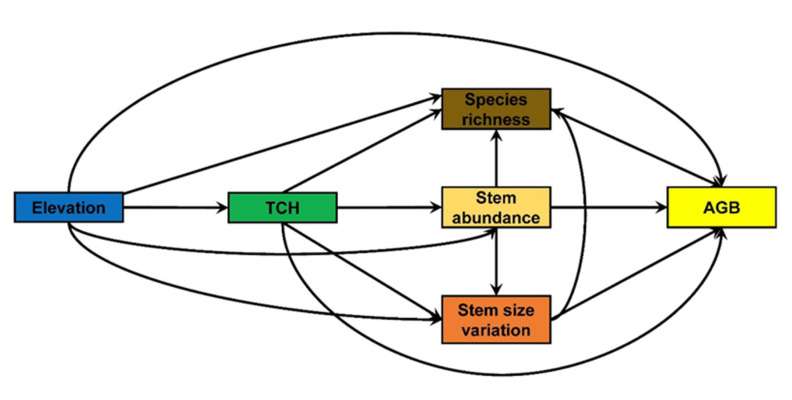
A conceptual model for linking elevation, mean top canopy height (TCH), stem size variation, stem abundance, species richness and aboveground biomass (AGB) across spatial scales in tropical forests.

**Figure 2 plants-12-01343-f002:**
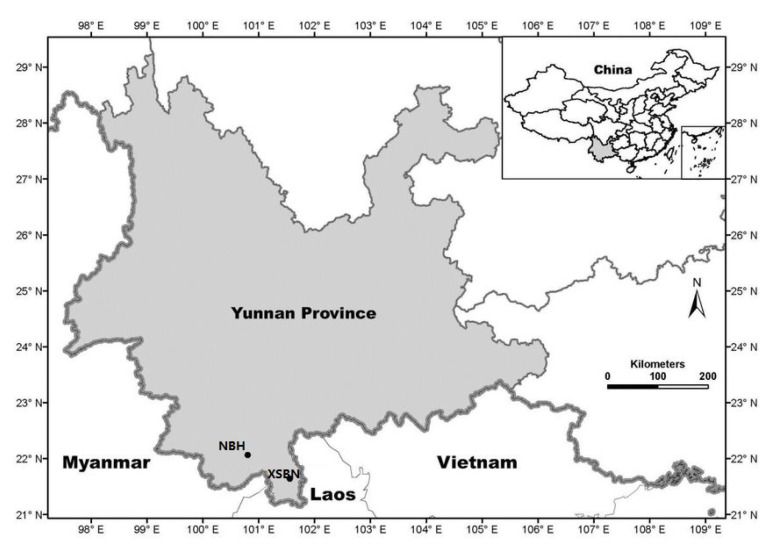
Location of the study area and the sampling points. NBH and XSBN stand for Nabanhe plot and Xishuangbanna plot, respectively.

**Figure 3 plants-12-01343-f003:**
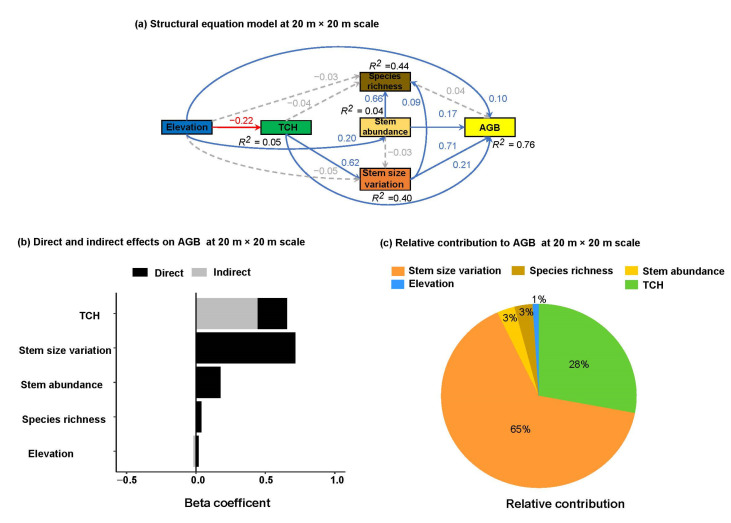
Structural equation model (**a**) for linking elevation, TCH, stem size variation, stem abundance, species richness and AGB at a small scale (i.e., 20 m × 20 m). Blue and red arrows represent significant positive and negative paths, respectively (*p* < 0.05) whereas dashed arrows show non-significant paths (*p* > 0.05). For each path, a standardized regression coefficient is shown. R^2^ indicates the total variation in a dependent variable, which is explained by the combined independent variables. See Appendix A for statistics. CFI = 1.000, GFI = 1.000, SRMR = 0.004, Chi-square = 0.190, *p*-value = 0.663. (**b**) Comparison of direct (dark bars) and indirect (grey bars) effects, derived from structural equation model, of elevation, TCH, stem size variation, stem abundance, and species richness on AGB. (**c**) Relative contributions of elevation, TCH, stem size variation, stem abundance, and species richness on AGB.

**Figure 4 plants-12-01343-f004:**
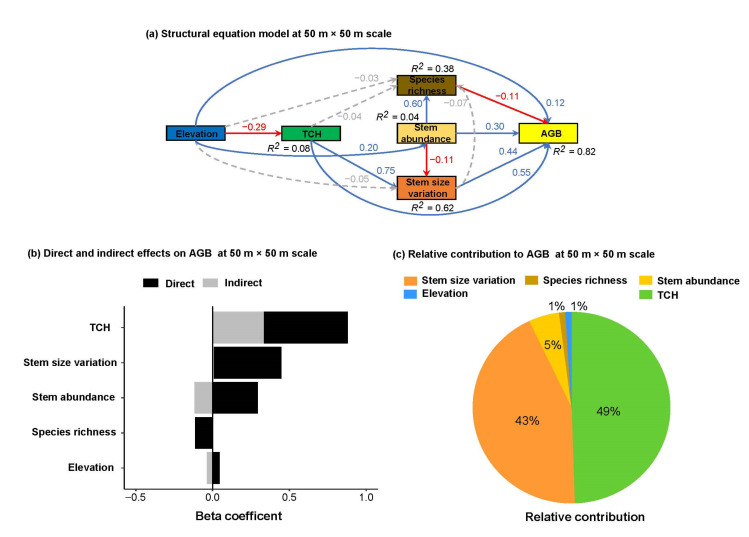
Structural equation model (**a**) for linking elevation, TCH, stem size variation, stem abundance, species richness and AGB at a large scale (i.e., 50 m × 50 m). Blue and red arrows represent significant positive and negative paths, respectively (*p* < 0.05) whereas dashed arrows show non-significant paths (*p* > 0.05). For each path, a standardized regression coefficient is shown. R^2^ indicates the total variation in a dependent variable, which is explained by the combined independent variables. See Appendix A for statistics. CFI = 0.997, GFI = 0.998, SRMR = 0.04, Chi-square = 2.292, *p*-value = 0.130. (**b**) Comparison of direct (dark bars) and indirect (grey bars) effects, derived from structural equation model, of elevation, TCH, stem size variation, stem abundance, and species richness on AGB. (**c**) Relative contributions of elevation, TCH, stem size variation, stem abundance, and species richness on AGB.

## Data Availability

The tree census data used in this study can be accessed by contacting the plot PI through ForestGEO Data Portal website (http://ctfs.si.edu/datarequest/).

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
