# Peer review of "Top Canopy Height and Stem Size Variation Enhance Aboveground Biomass across Spatial Scales in Seasonal Tropical Forests"

_plants, 2023, doi:10.3390/plants12061343_

Round 1

Reviewer 1 Report

Dear Authors,

I have never before read a paper where the Results and Discussion are before the Methods; I strongly suggest that you restructure to a more conventional order.

The lines 283 to 285 read to me like a comment from an earlier reviewer that has accidentally been included into the text.

I would add a little bit about the importance of Parahorea chinese as your dominant tree in terms of AGB but also that it a vulnerable species.

I would not expect species diversity on it's own to be important but the proportion of pioneer species to shade tolerant species could be?

I thought the discussion on errors and  uncertainty was weak, I would like to see more discussion on that.

It wasn't clear in the text how you went about constructing the SEM, which seems to me to be the innovative component of the paper. Did you use one of the packages in R? in which case you need to cite it.

Reviewer 2 Report

Dear authors, after reviewing your manuscript I consider that the research conducted is appropriate. However, there are two factors that prevent me from making a decision. If you remedy them, then I will be able to make a decision on your manuscript:

1.- The structure of the manuscript is not correct. The structure should be: introduction, methodology, results, discussion and conclusions. Additionally, there are aspects of the introduction that are methodological and should be eliminated.

2.- There is a great excess of self-citations in the manuscript. I understand that the authors have a long trajectory researching this field of knowledge, but self-citations should not exceed 10%. That is, if there are 68 references, there should only be a maximum of 7 references from the authors themselves, and there are many more in the current version of the manuscript. This calls into question the scientific quality of the manuscript, especially considering that other authors work in the same field. I recommend that the authors do a broader bibliographic search, eliminating their self-citations.

Kind regards,

Round 2

Reviewer 1 Report

The authors have answered all my comments very nicely. I would like to thank them for their response.

Author Response

Thanks a lot for your comments.

Reviewer 2 Report

My comments have not been adressed.

Round 3

Reviewer 2 Report

Dear authors, your manuscript has been improved. I attach some suggestions:

1.- It is highly recommended to attach a figure with the location of the study area and the sampling points.

2.- I think it would be better if, in methodology, you specify the standarization methods with equations.

Kind regards, 

Round 4

Reviewer 2 Report

My comments have been addressed. Kind regards,